# Intra- and Extrathoracic Malignant Tracheoesophageal Fistula—A Differentiated Reconstructive Algorithm

**DOI:** 10.3390/cancers13174329

**Published:** 2021-08-27

**Authors:** Thomas Kremer, Emre Gazyakan, Joachim T. Maurer, Katja Ott, Andreas Gerken, Marc Schmittner, Ulrich Ronellenfitsch, Ulrich Kneser, Kai Nowak

**Affiliations:** 1Burn Center, Department of Plastic and Hand Surgery, St. Georg Clinic, 04129 Leipzig, Germany; 2Department of Hand, Plastic and Reconstructive Surgery, Burn Center, BG Trauma Center Ludwigshafen, University Heidelberg, 67071 Ludwigshafen, Germany; emre.gazyakan@bgu-ludwigshafen.de (E.G.); ulrich.kneser@bgu-ludwigshafen.de (U.K.); 3Mannheim University Medical Center, Department of Ear, Nose and Throat Surgery, 68167 Mannheim, Germany; joachim.maurer@umm.de; 4Department of General, Vascular and Thoracic Surgery, RoMed Klinikum, 83022 Rosenheim, Germany; chirurgie-ro@romed.de (K.O.); kai.nowak@romed.de (K.N.); 5Mannheim University Medical Center, Department of Surgery, 68167 Mannheim, Germany; andreas.gerken@umm.de (A.G.); ulrich.ronellenfitsch@uk-halle.de (U.R.); 6Mannheim University Medical Center, Department of Anesthesia and Critical Care Medicine, 68167 Mannheim, Germany; marc.schmittner@ukb.de; 7Department of Visceral, Vascular and Endocrine Surgery, Martin-Luther-University Halle-Wittenberg, University Medical Center Halle (Saale), 06120 Halle, Germany

**Keywords:** tracheoesophageal fistula, complication management, free flaps, pedicled flaps, chimeric flaps

## Abstract

**Simple Summary:**

Tracheoesophageal fistulae (TEF) after oncologic resections represent a therapeutic challenge. Reconstructive options vary from the upper to the lower airways and include the complete therapeutic armamentarium from pedicled flaps to chimeric free flaps. Derived from the experience of 18 oncologic patients with TEF, we present a therapeutic algorithm that may guide future treatment strategies and shows that an interdisciplinary approach leads to satisfying success rates. However, disease-specific morbidity has to be anticipated.

**Abstract:**

Background: Tracheoesophageal fistulae (TEF) after oncologic resections and multimodal treatment are life-threatening and surgically challenging. Radiation and prior procedures hamper wound healing and lead to high complication rates. We present an interdisciplinary algorithm for the treatment of TEF derived from the therapy of consecutive patients. Patients and methods: 18 patients (3 females, 15 males) treated for TEF from January 2015 to July 2017 were included. Two patients were treated palliatively, whereas reconstructions were attempted in 16 cases undergoing 24 procedures. Discontinuity resection and secondary gastric pull-up were performed in two patients. Pedicled reconstructions were pectoralis major (*n* = 2), sternocleidomastoid muscle (*n* = 2), latissimus dorsi (*n* = 1) or intercostal muscle (ICM, *n* = 7) flaps. Free flaps were anterolateral thigh (ALT, *n* = 4), combined anterolateral thigh/anteromedial thigh (ALT/AMT, *n* = 1), jejunum (*n* = 3) or combined ALT–jejunum flaps (*n* = 2). Results: Regarding all 18 patients, 11 of 16 reconstructive attempts were primarily successful (61%), whereas long-term success after multiple procedures was possible in 83% (*n* = 15). The 30-day survival was 89%. Derived from the experience, patients were divided into three subgroups (extrathoracic, cervicothoracic, intrathroracic TEF) and a treatment algorithm was developed. Primary reconstructions for extra- and cervicothoracic TEF were pedicled flaps, whereas free flaps were used in recurrent or persistent cases. Pedicled ICM flaps were mostly used for intrathoracic TEF. Conclusion: TEF after multimodal tumor treatment require concerted interdisciplinary efforts for successful reconstruction. We describe a differentiated reconstructive approach including multiple reconstructive techniques from pedicled to chimeric ALT/jejunum flaps. Hereby, successful reconstructions are mostly possible. However, disease and patient-specific morbidity has to be anticipated and requires further interdisciplinary management.

## 1. Introduction

Multimodal treatment options including chemo- and immunotherapy as well as radiotherapy and surgery are usually applied in patients suffering from head and neck cancers as well as esophageal malignancies [1]. Significant survival rates can be achieved even in recurrent cases [2,3,4,5,6,7]. However, this multimodal therapy regimen and especially radiotherapy can lead to long-term side effects including tracheoesophageal or enterocutaneous fistulae [1,8]. Tracheoesophageal fistulae (TEF) develop in untreated esophageal cancer in 5% to 15% of cases, whereat chemoradiotherapy triggers fistulation [9]. Furthermore, prior radiotherapy renders spontaneous healing unlikely due to the diminished healing capacities of radiated tissue. On the other hand, morbidity rates of patients suffering from TEF are significant and result from chronic aspiration leading to pneumonia and therapy-resistant cough, bleeding, and esophageal stenosis or occlusion with dysphagia as well as impaired vocal rehabilitation [10,11]. Non-resolving aspiration pneumonia may lead to pulmonary sepsis with fatal consequences within 6 to 12 weeks [9]. A prophylactic therapeutic option in patients with advanced carcinomas and after radiation therapy may be the use of salivary Montgomery prostheses in order to prevent fistula formation [12].

Surgically, the application of vascularized, non-radiated tissue to close the fistulae and to reconstruct the airway as well as the digestive tract is frequently required. Potential reconstructive options include pedicled flaps such as the pectoralis major [13,14] or latissimus dorsi flap [15,16], perforator flaps [17] or local muscle flaps such as intercostal muscle (ICM) [18], serratus anterior muscle [16] or pedicled pericard patches [19]. Other potential flaps include the free microvascular radial forearm flap, the anterolateral thigh (ALT) flap or the free jejunal flap [14,20,21,22,23]. These flaps allow the transposition of well vascularized tissue to improve wound healing capacities [24]. However, the underlying conditions with poor tissue quality and reduced overall health status of the patients still lead to considerable perioperative morbidity rates at the recipient site as well as donor site complications [14]. The treatment of these complications can be life-threatening by itself and frequently requires multiple disciplines to be involved in the management.

Reconstructive decision-making in this difficult situation depends on defect size and localization, the possibility of complete tumor resection, the patient’s overall health status, potential donor sites, the accessibility of recipient vessels and tissue components that have to be addressed. Therefore, an individualized approach for each patient has to be planned. Several surgical teams such as ENT, Maxillofacial Surgery, Visceral and Thoracic Surgery as well as Plastic Surgery have to be involved. This leads to different therapeutic experiences and toolboxes guiding treatment decisions. Interdisciplinary case discussion together with a dedicated anesthesiologist experienced in extraordinary airway management strategies intra- and postoperatively is essential for good results. Analyzing our patients as well as peri- and intraoperative experiences, we developed an interdisciplinary algorithm that may guide future decision-making according to defect localization, defect size and primary versus recurrent TEF.

## 2. Patients and Methods

### 2.1. Patients

The study was approved by the local ethics committee (No: 2015-854R-MA). Eighteen patients treated for TEF in our institutions between January 2015 and July 2017 were included in this retrospective analysis. The mean age at surgery was 67.8 (+/− 6.4) years. Three women and 15 men were included in the study. Patients suffered from esophageal cancers (*n* = 11) and malignancies of the larynx (*n* = 3), pharynx (*n* = 1) or hypopharynx (*n* = 3). The head and neck patients all underwent laryngectomy and chemoradiotherapy and developed TEF secondarily. Two patients with lesions at the cervicothoracic junction suffered from TEF directly postoperatively, whereas the others (*n* = 2) developed late TEF. Intrathoracic TEF were predominantly late lesions (*n* = 4). One patient suffered from TEF due to anastomotic failure postoperatively. In 16 patients, a curative oncological approach was followed. Except one, all patients had prior neoadjuvant or definitive chemoradiotherapy. Clinically, the patients were divided into three subgroups: 9 patients suffered from cervical fistula proximal from the jugulum, 4 patients were treated for fistulae at the cervico-thoracic junction, whereas another 5 patients had developed intrathoracic aerodigestive fistulae of the distal trachea or the central bronchi (Table 1).

### 2.2. Preoperative Evaluation

Firstly, patients with TEF were evaluated systemically for potential metastasis of the underlying oncologic disease. All patients underwent computed tomography of the chest as well as the abdomen. Positron emission tomography (PET) was applied when recommended by the multidisciplinary tumor board.

Secondly, the local extent of the fistula and/or potential esophageal stenosis was assessed. Here, contrast-enhanced MRI scans were performed to evaluate local recurrence. Additionally, a bronchoscopy was undertaken to examine the airways and to localize the fistula. The digestive tract was evaluated using pharyngoscopy, laryngoscopy or esophago-gastro-duodenoscopy according to the localization of the fistula that was observed radiologically. In patients with (sub-) total stenosis of the esophagus that could not be passed with the endoscope, an additional endoscopy was performed through the percutaneous gastrostomy tube to assess the distal extent of the stenosis or the fistula.

Additionally, CT-angiographies of the local vessels (neck, chest) were performed to define recipient vessels and—in selected cases—to evaluate pedicle vessels of potential flap donor sites, when a free flap was required.

### 2.3. Intra- and Postoperative Airway Management

Airway management during or after airway reconstruction is challenging. Planning and rescue strategies were discussed prior to surgery. Central airway fistulae and fistulation at the cervico-thoracic junction are most challenging as they cannot be excluded from ventilation by a simple tracheostomy as a cervical fistula. For these challenging lesions for both reconstruction and healing of a flap, the pressure of cuffs in the area of the fistula must be avoided. Furthermore, the clearance of bronchial secretions by bronchoscopy supports the healing of flaps. Prolonged side selective ventilation was performed in patients with bronchial fistulation to facilitate flap healing [25,26,27].

### 2.4. Operative Approach

Twenty-four reconstructive procedures were performed in 16 patients. Two patients suffering from very large TEF from esophageal cancer were not subjected to any reconstructive attempt. Of these, one patient underwent a distal closure of the esophagus and a palliative tracheostomy, whereas the other was irresectable and received best supportive care. For reconstructions, a free jejunum (*n* = 2), a pectoralis major (*n* = 2), an ALT flap (*n* = 2), a flow-through ALT flap combined with a free jejunum (*n* = 2), a sternocleidomastoid flap (*n* = 1) or an intercostal muscle flap (*n* = 6) was performed primarily. In two patients, a two-stage reconstruction was planned with initial esophageal discontinuity resection and closure of the bronchus using an intercostal muscle flap, and secondary gastric pull-up. Five patients required more than one procedure for reconstruction (average number of procedures: 2.2). Here, one free jejunum flap was performed after insufficient reconstruction using a pectoralis major flap (PM). Another patient after PM reconstruction required a second reconstructive attempt using an ALT flap. Unfortunately, this flap was also lost due to persistent infection before TEF-closure was finally performed by a combined ALT and anteromedial thigh (AMT) flap. One more case after discontinuity resection and ICM flap reconstruction needed an additional sternocleidomastoid flap before a gastric pull-up procedure was performed. Two intercostal muscle flaps failed to sufficiently close the aerodigestive fistula. A latissimus dorsi flap and a free ALT flap were secondarily applied in these cases.

### 2.5. Recipient Vessels in Free Flap Reconstructions

One free jejunum segment was anastomosed to the superior thyroid artery and to the jugular vein. All other free microvascular tissue transplantations were connected to the internal mammary artery (IMA) and vein (IMV) under the second rib. In two cases, a free chimera flap (ALT/jejunum) was performed. In these cases, the anastomosis of the ALT flap was performed to the IMA and IMV, respectively. The free microvascular jejunum was then anastomosed to the outflow of the ALT pedicle.

### 2.6. Intraoperative Perfusion Control of Flaps

Indocyanine green angiography was used to ensure and control vascular perfusion of free and pedicled flaps as well as gastrointestinal anastomoses [28,29,30]. The technology can be helpful to ensure the perfusion of pedicled flaps such as the ICM or before and after free jejunal transposition (Figure 1).

### 2.7. Outcome Measurements

Patients were evaluated for the success of the reconstruction (wound healing and fistula closure), the necessity of secondary reconstructive procedures and for 30-day morbidity and mortality rates.

## 3. Results

In 16 patients who underwent a reconstructive attempt, 11 reconstructions were primarily successful (61% of all 18 patients). Four out of seven remaining patients could be successfully treated after multiple reconstructive attempts, leading to an overall success rate of 83% TEF (15/18 cases). Thirty-day survival was achieved in 16 patients (89%). The initial TEF were associated with preoperative radiation therapy in 17 cases (95%). Oral ingestion was possible in 11 patients, two patients were able to swallow fluids and saliva but no solid food and three patients could not ingest orally (missing information in two patients).

One 60-year-old patient with cervical TEF after pharyngolaryngectomy and chemoradiotherapy was initially treated using a pedicled pectoralis major muscle flap. Despite the fact that the muscle flap was vital, wound healing was not achieved and the fistula persisted. Consequently, a second reconstruction using a free jejunum segment was performed. However, the postoperative course was still complicated for two reasons. First, the skin could not be closed over the jejunum and the remaining defect healed by secondary intention; second, a small pharyngocutaneous fistula persisted and required an epithesis. This patient suffered from distant metastases and therefore did not want further reconstructive procedures (see Figure 2).

Another 70-year-old male suffering from esophageal cancer was treated for intrathoracic TEF using an intercostal muscle flap. A persistent tracheoesophageal fistula was observed in the postoperative course. The patient was then treated by debridement and secondary reconstruction using a latissimus dorsi muscle flap to close the fistula. Although the fistula healed, the patient died during the postoperative course due to the preexisting aspiration pneumonia.

Three more patients required a second reconstruction after failure of the initial attempt (see above; for detailed information see Table 2).

### Therapeutic Algorithm Derived from the Experience

Analyzing the experiences from the case series, we developed a therapeutic algorithm to address TEF. The patients were divided into three groups: extrathoracic TEF, intrathoracic TEF and fistulae at the cervicothoracic junction. The first two groups were separated since we observed different therapeutic armamentaria to be required. TEF at the cervicothoracic junction were considered a separate entity because special therapeutic planning is required and technical difficulties were observed due to more difficult surgical exposure under the proximal sternum. Moreover, only 50% of reconstructions were successful.

Patients suffering from extrathoracic TEF were treated using pedicled flap reconstructions as a first-line treatment. Here, larger fistulae were reconstructed using a pectoralis major flap, whereas smaller lesions were planned for perforator flap reconstructions or a sternocleidomastoid turnover, when its vascularity was not compromised by bilateral neck dissections. Recurrent or persistent fistulae were usually treated using a free microvascular tissue transfer. In patients without esophageal stenosis, an ALT flap was performed to close the aerodigestive connection. If the patients additionally had a skin defect or a skin paucity (due to radiation therapy), a split ALT flap was performed because multiple perforators with separated skin islands allow the reconstruction of the digestive tract, the trachea and skin defects (see Figure 3). If patients suffered from proximal TEF with an additional stenosis or occlusion of the esophagus, patients were planned for a free jejunum segment transfer. Again, additional skin defects or a lack of neck skin proved to require a chimeric flap using an ALT flap combined with a free jejunum segment transfer (see Figure 4).

Patients with intrathoracic tracheoesophageal fistulae were planned for different approaches. Smaller lesions were primarily planned for reconstruction such as pedicled intercostal muscles or a vascularized pericardial flap, whereas larger lesions were either treated by a pedicled latissimus dorsi flap or discontinuity resection with a pedicled intercostal muscle to close the airway, and secondary gastric pull-up to reconstruct the digestive tract. In this algorithm (see Figure 1), free flaps would be performed when a latissimus dorsi flap would not be possible—even though this was not required in the presented series.

In some patients, a reconstructive attempt was not performed due to oncologic reasons or the patient’s overall health status—this is an important therapeutic decision that has to be included in the therapeutic decision-making process and algorithm (see. Figure 5).

## 4. Discussion

The reconstruction of tracheoesophageal fistulae (TEF) can be very challenging, especially in patients suffering from malignancies, because radiation therapy has frequently been applied. This has been shown to be a significant risk factor for fistula formation [31,32]. Non-surgical treatment options are mostly reserved for palliative care since significant complication rates are inevitable [20]. Surgical treatment options using local tissue rather than vascularized tissue transfers have been described using different suture techniques [33], but do also show high morbidity rates. Consequently, a transfer of vascularized, non-radiated tissue is frequently applied in these patients. The pectoralis major myocutaneous (PMMC) flap was introduced for head and neck reconstruction early and is still a workhorse for reconstruction [34,35]. Moreover, different pedicled flaps for intrathoracic TEF have been described, such as intercostal muscle flaps, a latissimus dorsi flap or a serratus anterior muscle flap [15,16,18]. These flaps are easy and safe treatment options. However, the rotation arc of each flap is limited and the border zone of pedicled flaps is not always reliable. This may be the reason why wound dehiscence is the most frequent problem observed after pedicled flap reconstruction, as described by Anehosur and colleagues in a recent study on 150 PMMC flaps [35]. On the contrary, free flaps have fewer limitations regarding rotation arc and geometry, and have therefore been increasingly applied. Here, the free radial forearm flap, the free jejunal segment flap and the anterolateral thigh (ALT) flap are most frequently performed [14,20,21,22,23]. A comparison of pedicled PMMC flaps versus fasciocutaneous free flaps showed no significant differences between the groups [13]. However, a review of the literature revealed that PMMC flaps had higher reported fistula rates (24.7 vs. 8.9%, *p* < 0.0001) and requirement for reoperation (11.3 vs. 5.5%, *p* = 0.04) than free fasciocutaneous flaps. The two groups showed comparable rates of flap loss or necrosis (3.5 vs. 2.0%, *p* = 0.32; [13]).

However, overall complication rates and perioperative morbidity in TEF patients are relatively high—regardless of the reconstructive choice. In fact, pedicled and free flaps in our opinion are just one tool in the therapeutic armamentarium and should be applied using a differentiated strategy including all therapeutic options. Therefore, we propose a new algorithm for all patients suffering from TEF from the cricoid to the carina. This algorithm also includes the option of “no-reconstruction”, whenever the physical status of the patient or the underlying disease do not allow extended surgical treatment. In these patients, the focus should be to do no harm and the relatively high perioperative risk has to be weighed up against the potential benefit.

Whenever patients are candidates for reconstruction, the proposed algorithm includes the TEF localization as one important decisive fact. We observed that extrathoracic TEF require a different set of flaps in comparison to intrathoracic lesions. Another basis for decision-making is primary versus recurrent TEF, the size of the lesion and a potential additional skin paucity in extrathoracic cases. A special subgroup of patients in our experience were TEF at the cervico-thoracic junction, because here extrathoracic as well as intrathoracic reconstructive options may be possible, but specific intraoperative difficulties have to be anticipated due to problematic surgical exposure and uncertain suture possibilities. The algorithm that is derived from our experience may facilitate an easier and structured approach in future patients. To the best of our knowledge, no comparable approach has been described to date. However, similar reconstructive algorithms have been described for patients after pharyngolaryngectomy [14,36]. These algorithms mainly include the ALT, radial forearm and free or supercharged jejunum flap, but also use pedicled flaps such as the pectoralis major, the supraclavicular flap or even a free vertical rectus abdominis myocutaneous flap. The authors also describe that one important fact is the availability of neck skin. If skin is rare and neck resurfacing is required, flexible flap combinations are recommended by the authors [36].

With regard to the high spontaneous short-term mortality of patients suffering from TEF [9], the 30-day mortality of 5.5% in the operated patients in our series is considerably low. Perioperative morbidity (24%) and mortality (*n* = 2; 12%) in our series are comparable to other studies. Selber and colleagues describe a frequency of overall complications of almost 40%, and of complications at the recipient site of 23% [14]. Other series comparing different techniques describe an incidence of recurrent fistulae of 11% and 22% after pharyngoesophageal reconstruction using a PMMC flap or a free fasciocutaneous flap, respectively. Additionally, the literature reviewed in this study revealed a 25% incidence of recurrent fistulae after PMMC reconstructions, whereas an incidence of 9% was described after fasciocutaneous free flaps [13]. In our series, we included two PMMC flaps that both failed and required free flaps for reconstruction. However, we still included pedicled PMMC flaps in the treatment algorithm as a first-line option since we have good experiences using this flap in other cancer cases, the number of patients is low and the literature does not show a clear trend towards or against pedicled versus free flaps.

Perez-Smith and colleagues describe leakage rates of 17% in their series using a free jejunum flap, compared to 19% after free fasciocutaneous flaps [22]. Lower morbidity is described by others using different flaps in selected cases and other localizations. Mirghani et al. observed an 8% complication rate using an internal mammary artery perforator flap in a small series of 12 patients [8]. The reconstruction of intrathoracic tracheoesophageal defects using extrathoracic muscle flaps leads to 8.2% morbidity and 10% mortality, respectively.

Sharaf and colleagues compared pharyngoesophageal reconstructions with and without additional skin reconstructions using a second ALT skin island on an additional perforator, an ALT/AMT flap or an ALT with additional vastus lateralis muscle. In some cases, an additional PMMC flap, a supraclavicular flap or a second free flap were performed. Interestingly, this group showed a lower fistula rate when neck resurfacing was performed [36]. Likewise, Moradi and colleagues changed their protocol from free jejunum segment reconstruction for pharyngeal defects to a free jejunum segment together with PMMC flaps for soft tissue augmentation after a fatal carotid blowout due to persistent infection [21]. This group also observed a reduction in terms of fistula formation and wound dehiscence when the soft tissues of the neck were additionally addressed.

Likewise, we observed wound healing problems in our patients when wound closure was impossible or insufficient due to skin paucity after reconstruction using a free jejunum segment. A second important difficulty may be poor recipient vessels in patients suffering from TEF. In fact, a vessel-depleted neck is frequently observed in these patients due to repetitive surgery such as oncologic resection including neck dissections and radiation. Here, alternative vessels outside the field of radiation and away from previous procedures can offer a safer option. In these cases, one reconstructive alternative solving the problem of a lack of skin as well as poor recipient vessels can be flow-through free flap combinations. In this context, Ciudad and colleagues described the free radial forearm flap as a “vascular bridge” in head and neck reconstruction. Here, this group anastomosed a free radial forearm flap to the internal mammary or the thoracoacromial vessels outside the field of radiation and performed a second anastomosis to the distal radial artery to create a chimeric flap. The radial forearm flaps were combined with ALT flaps, free jejunum segment flaps or a free osseous fibula [37]. The ALT flap itself has been described as a potential flow-through flap for sequential connections of multiple flaps [38]. Consequently, we changed our approach in patients with free jejunum segment flaps and insufficient soft tissue of the neck towards chimeric ALT flaps that were anastomosed to internal mammary vessels outside prior operation and radiation fields, and that served as a “vascular bridge” for the jejunum, which was anastomosed to the distal ALT pedicle. Hereby, the two problems of scarce neck skin and poor recipient vessels could be solved in one approach. To our knowledge, the ALT/jejunum combination with a common blood supply for head and neck reconstruction has not been described before.

In our experience, the internal mammary artery and vein are safe donor and recipient vessels for free flaps, when flaps with a sufficient pedicle length are applied. The main advantage is that these vessels are usually outside prior surgical sites or radiation fields. However, special contraindications such as previous arterial cardiac bypasses and thoracic surgeries have to be considered.

In summary, reconstructive procedures in TEF patients require meticulous interdisciplinary planning and also a shared intraoperative approach. This interdisciplinary concerted action including anesthesiologists, ENT, abdominal as well as plastic surgeons has been described by others [22] and was likewise performed in our series. It is not only required intraoperatively, but is of utmost importance for postoperative care and complication management, including perfusion control, restitution of swallowing and speech, and potential endoscopic controls and interventions. This applies especially to patients suffering from TEF located between the cricoid and the carina.

Limitations of the presented series are the retrospective design and the relatively small sample size that does not allow statistic comparisons between different types of reconstructions or localizations. Additionally, only patients that were presented to our team for potential reconstructions were included. Therefore, the group of patients that would be primarily subjected to palliative treatment may be underestimated.

## 5. Conclusions

Malignant TEF after complicated oncologic resection and multimodal treatment including radiation require concerted interdisciplinary efforts for successful reconstruction. Derived from our experience, we propose a consensus algorithm that may be applicable to all patients suffering from malignant TEF from the cricoid to the carina. This algorithm comprises TEF localization, size, recurrence and skin availability for operative planning and includes multiple reconstructive options from pedicled flaps to chimeric ALT/jejunum flaps. Hereby, the majority of patients can be successfully treated. However, disease and patient-specific morbidity has to be anticipated and requires further interdisciplinary management.

## Figures and Tables

**Figure 1 cancers-13-04329-f001:**
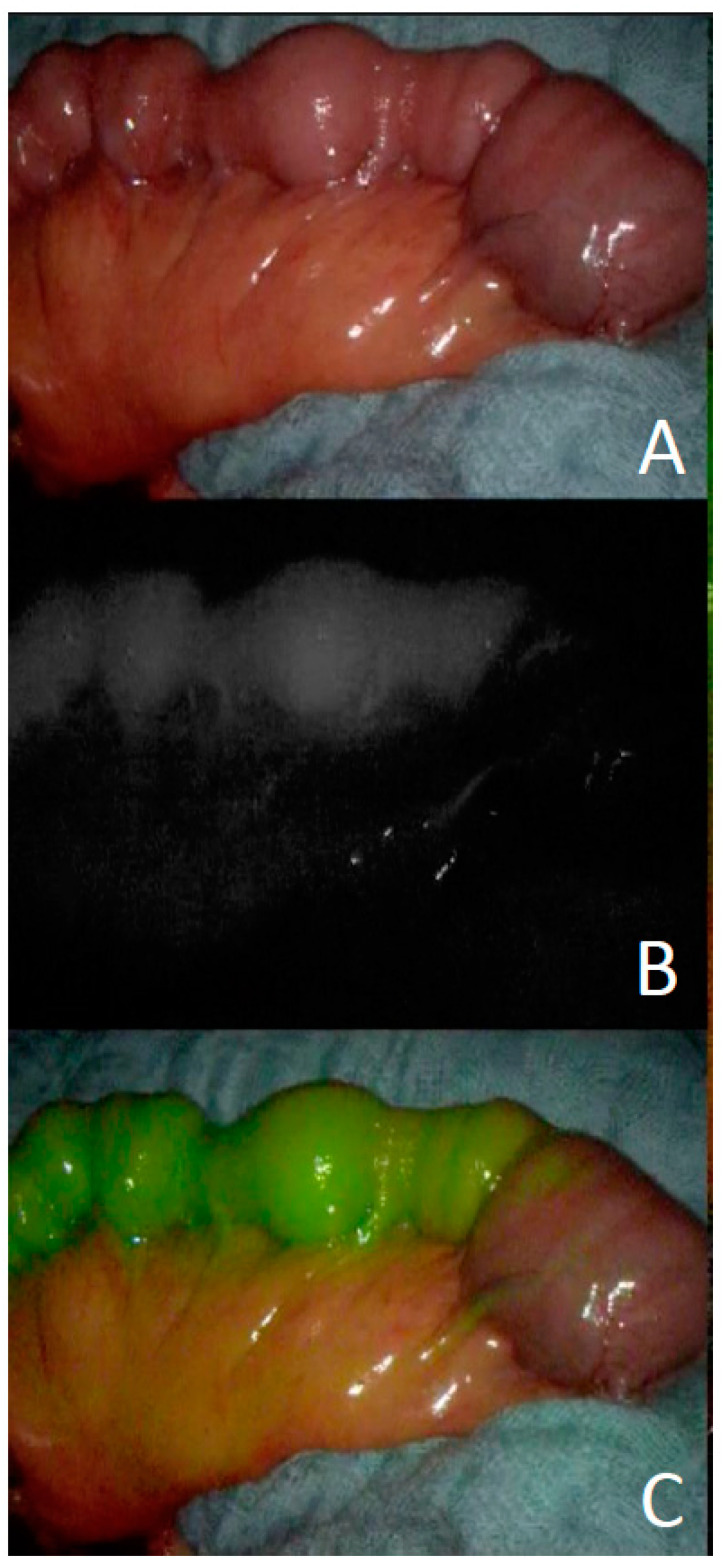
Intraoperative imaging using indocyanine green after isolation of a jejunum segment ((**A**) native view; (**B**) ICG-Perfusion; (**C**) merged view).

**Figure 2 cancers-13-04329-f002:**
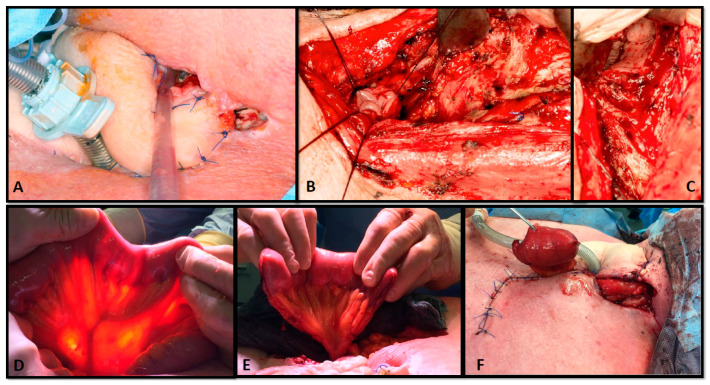
Sixty-year-old patient after reconstruction for a pharyngotracheal fistula after pharyngolaryngectomy and chemoradiotherapy, which was initially treated using a pedicled pectoralis major muscle flap. A second reconstruction was required for persistent TEF (**A**). After debridement and dissection of the distal esophagus (**B**) and the proximal pharynx ((**C**) arrow), reconstruction was planned using a free jejunum segment (**D**,**E**) that was anastomosed to the internal mammary vessels. The skin could not be closed and was left for healing by secondary intention (**F**). A remaining proximal pharyngocutaneous fistula was closed using an epithesis since the patient denied further operations.

**Figure 3 cancers-13-04329-f003:**
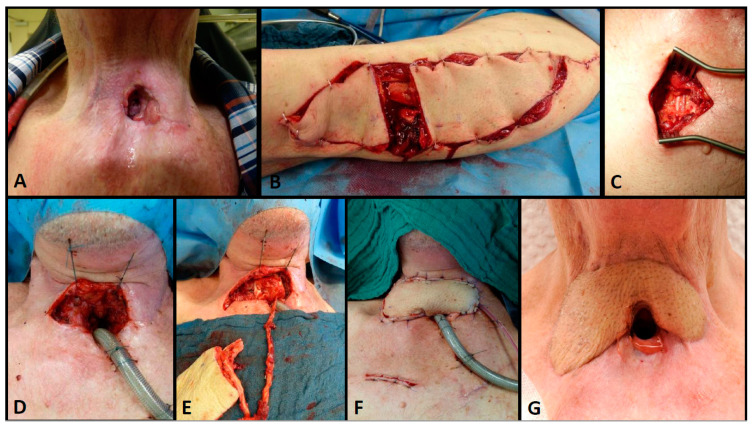
Seventy-four-year-old patient who had undergone laryngectomy, neck dissection and radiotherapy for laryngeal cancer. He was treated for an extrathoracic TEF (**A**) using a split anterolateral thigh (ALT) flap with two skin islands (**B**) that was anastomosed to the internal mammary vessels (**C**). One skin island was used for fistula closure (**D**,**E**) and the other to cover the skin defect cranial from a permanent tracheostomy (**F**,**G**); long term result after 6 months.

**Figure 4 cancers-13-04329-f004:**
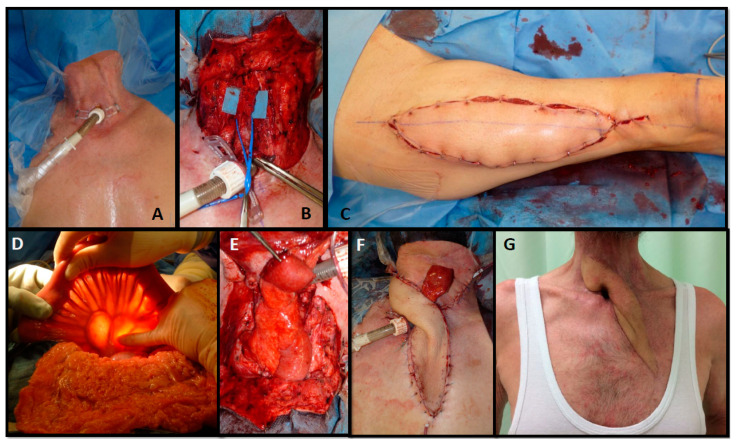
Sixty-six-year-old male suffering from a pharyngotracheal fistula after pharyngolaryngectomy (**A**). The esophagus was occluded over 7 cm due to a post-radiotherapy stenosis and the patient was fed via a percutaneous gastrostomy tube for 7 years (**B**); intraoperative view of the esophagus remnant. Reconstruction was performed using an ALT flow-through flap (**C**). The digestive tract was reconstructed using a free jejunum segment (**D**); intraoperative view to select the jejunum segment (**E**); intraoperative view after inset that was anastomosed distally to the outflow of the ALT (**F**); intraoperative view after wound closure; a segment of the jejunum was separated for postoperative perfusion monitoring (**G**); long-term result; the patient was eating and drinking normally.

**Figure 5 cancers-13-04329-f005:**
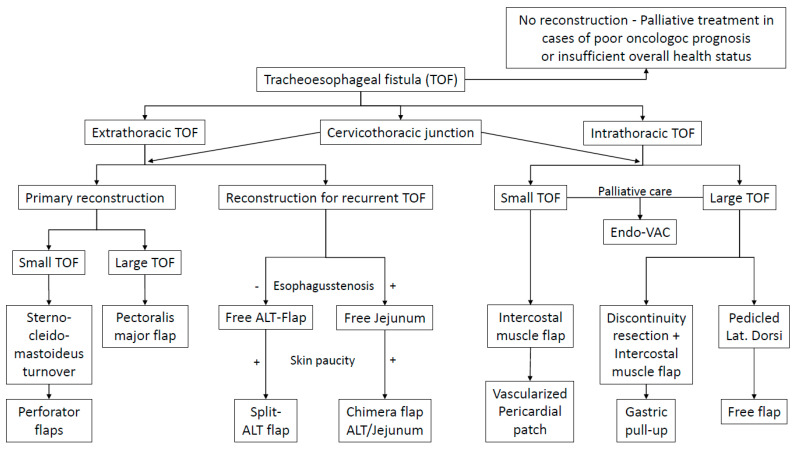
Treatment algorithm for the treatment of patients suffering from tracheoesophageal fistulae.

**Table 1 cancers-13-04329-t001:** Description of the patient cohort treated for TEF.

		N	Percent
Age (Years)		67.1 ± 7.0	
Sex	Female	3	17%
	Male	15	83%
Tumor location	Esophagus	11	61%
	Larynx	3	17%
	Pharynx	1	5%
	Hypopharynx	3	17%
Fistula classification	Cervical	9	50%
	Cervico-thoracic	4	22%
	Intrathoracic	5	28%
Radiation	Yes	17	95%
	No	1	5%

**Table 2 cancers-13-04329-t002:** Outcomes after reconstruction for TEF (ALT, anterolateral thigh flap; AMT, anteromedial thigh flap; sternocleido, sternocleidomastoid muscle flap; Discont. Resection, discontinuity resection of the esophagus; TEF, tracheoesophageal fistulae; ICM, intercostal muscle flap; Recon, reconstruction).

Patient-No.	Localization of TEF	Malignancy	1st Recon.	2nd Recon.	3rd Recon.	Success of Recon.	30-Day Survival
1	Cervical	Esophageal	Jejunum	-	-	Yes	Yes
2		Laryngeal	Jejunum	-	-	Yes	Yes
3		Esophageal	None	-	-	No	No
4		Laryngeal	PM	Jejunum	-	Yes	Yes
5		Hypopharyngeal	Split-ALT	-	-	Yes	Yes
6		Laryngeal	Split-ALT	-	-	Yes	Yes
7		Hypopharyngeal	PM	ALT	ALT/AMT	Yes	Yes
8		Pharyngeal	ALT/Jejunum	-	-	Yes	Yes
9		Hypopharyngeal	ALT/Jejunum	-	-	Yes	Yes
10	Cervico-thoracic	Esophageal	Discont. Resection Sterno-cleido.	-	-	Yes	Yes
11		Esophageal	ICM	ALT	-	No	Yes
12		Esophageal	Discont. Resection ICM	Sterno-cleido.	Gastric pull-up	Yes	Yes
13		Esophageal	None	-	-	No	Yes
14	Intrathoracic	Esophageal	Discont. Resection ICM	Gastric pull-up		Yes	Yes
15		Esophageal	ICM	LD	-	Yes	No
16		Esophageal	ICM	-	-	Yes	Yes
17		Esophageal	ICM	-	-	Yes	Yes
18		Esophageal	ICM	-	-	Yes	Yes

## Data Availability

The data presented in this study are available on request from the corresponding author.

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
