# Peer review of "Intra- and Extrathoracic Malignant Tracheoesophageal Fistula—A Differentiated Reconstructive Algorithm"

_cancers, 2021, doi:10.3390/cancers13174329_

Round 1
Reviewer 1 Report
1. The introduction can be significantly shortened.
2. You state: "16 patients who underwent a reconstructive attempt, 11 reconstructions were primarily successful (61% of all 18 patients). Long-term success regarding closure of the TOF after repetitive reconstructions was achieved in 15 of 18 cases (83%)." I don't feel this statement is correct as you include 11 patients that healed after primary surgery in the latter group. Long term success rate after repetitive recon is 4/7 in my opinion.
Author Response
We thank the reviewer for the important recommendations. We modified the manuscript accordingly:
- The introduction can be significantly shortened.
=> the introduction was significantly shortened
- You state: "16 patients who underwent a reconstructive attempt, 11 reconstructions were primarily successful (61% of all 18 patients). Long-term success regarding closure of the TOF after repetitive reconstructions was achieved in 15 of 18 cases (83%)." I don't feel this statement is correct as you include 11 patients that healed after primary surgery in the latter group. Long term success rate after repetitive recon is 4/7 in my opinion.
=> We agree, that our statement may have been misleading. Therefore, we stated the facts more precisely.
"In 16 patients who underwent a reconstructive attempt, 11 reconstructions were primarily successful (61% of all 18 patients). 4 out of 7 remaining patients could be successfully treated after multiple reconstructive attempts, leading to an overall success rate of 83% (15/18 cases)."
Reviewer 2 Report
The authors present the results of a retrospective cohort of 18 patients treated for Tracheo-esophageal Fistula in time frame2015-2017. They try to introduce a therapeutic algorithm taking into account the complexity of this pathology
Authors prospect a paper on one of the most challenge topic in oncologic surgery . they provide good results on a reasonable size cohort, I would recommended minor revision of the paper to be suitable for a publication.
Here some comments for each section.
Title
“tracheoosophagea”, typing error, please correct it (esophagus or oesophagus). If authors use esophagus please change even abbreviation from TOF to TEF
Abstract
please add the time frame of the study in this section
Lines 40,41: the title says “reconstructive”, but two patients were treated palliative without any kind of reconstruction so these two patient should be removed from the study.
Line 47: Authors state that 11 patients resolved on first attempt were primarily successful but in table 2 is possible to count only 10. Please explain it or correct it.
Introduction Well done
Patients and methods
Not clear if the fistula is presented in the early post-op (after tumor ablation) or late after CRT. Or a group have early fistula and another one late fistula. Please explain
Line 116: a list of sites of the head and neck tumor (larynx, pharynx and hypopharynx) is not so useful, would be better to classify
patient related to surgery (e.g. total laryngectomy or extended TL, primary closure of the pharynx or reconstruction etc)
line 150-151: these 2 patients didn't receive any reconstruction, for this reason should be removed from the study
line 155: about sternocleoidomastoidues flap, in which case has been used this flap? Is patient who received neck dissection? During neck dissection all pedicles of the muscle are resected and the it could result not vascularized, in these cases the procedure could become extremely hazardous (please explain this risk).
Line 159-161: “Another failure after pectoralis major reconstruction was finally salvaged using a combined ALT and anteromedial thigh (AMT) flap after a free ALT flap was lost due to persistent infection” this sentence is not clear and could be better built (did this patient receive 3 procedures?)
Punctuation in the legends is not correct, commas are used instead of semicolons and vice versa Line 86: reference 6 is not appropriated, please provide adequate reference.
Line 231: Authors state that first line in extrathoracic fistula their policy is to use pedicle flap, but all pedicle flap in cervical or cervical-thoracic fistula required second or third stage. How do the authors explain this fact, is radiotherapy much more impacting in head and neck area? under this
light would be wiser to use free flap upfront in this area when possible (not in vessels depleted neck), add a sentence or in the algorithm please.
Line 253, 255 brackets not in the right place
Line 262: Author included, even in the algorithm, a vascularized pericardial flap, but they didn’t use in this series. Please remove at list from the algorithm
Discussion Well done
References Fine
Author Response
Title
“tracheoosophagea”, typing error, please correct it (esophagus or oesophagus). If authors use esophagus please change even abbreviation from TOF to TEF
=> Thank you for this important comment. We corrected the title and changed TOF to TEF in the complete manuscript.
Abstract
please add the time frame of the study in this section
=> we added the timeframe to the abstract
"18 patients (3 females, 15 males) treated for TEF from 1/2015 to 7/2017 were included."
Lines 40,41: the title says “reconstructive”, but two patients were treated palliative without any kind of reconstruction so these two patient should be removed from the study.
=> Prior to submission, we discussed this topic intensively. We believe, that the option of "no treatment" should always be considered and is therefore an important part in any reconstructive algorithm. Therefore, we included this option in our treatment algorithm and would prefer to present these 2 patients in the manuscript. We highlighted this option in the discussion section to explain, why these two patients were included in the study .
"This algorithm also includes the option of “no-reconstruction”, whenever the physical status of the patient or the underlying disease do not allow extended surgical treatment. In these patients, the focus should be to do no harm and the relatively high perioperative risk has to be weighed up against the potential benefit."
Line 47: Authors state that 11 patients resolved on first attempt were primarily successful but in table 2 is possible to count only 10. Please explain it or correct it.
=> patient 14 was treated in two planned stages. Closure of the TOF was already achieved using the ICM in the discontinuity resection procedure. We judged this patient as primarily sucessful regarding TEF-closure. Therefore, 11 is correct.
Patients and methods
Not clear if the fistula is presented in the early post-op (after tumor ablation) or late after CRT. Or a group have early fistula and another one late fistula. Please explain.
- This is really an important information that hast o be included in the manuscript. It was added tot he manuscript
- “The head and neck patients all underwent laryngectomy and chemoradiotherapy and developed TEF secondarily. 2 patients (No. 10 and 11, table 2) with lesions at the cervicothoracic junction suffered from TEF direct postoperatively, whereas the others (n=2) developed late TEF. Intrathoracic TEF were predominantly late lesions (n=4). One patient suffered from TEF due to anastomotic failure postoperatively (patient 14, table 2)”.
Line 116: a list of sites of the head and neck tumor (larynx, pharynx and hypopharynx) is not so useful, would be better to classify; patient related to surgery (e.g. total laryngectomy or extended TL, primary closure of the pharynx or reconstruction etc)
- All head and neck cases had TEF after laryngectomies and tracheostomy. We added this informmation to the manuscript
Patients suffered from esophageal cancers (n=11) and malignancies of the larynx (n=3), pharynx (n=1) or hypopharynx (n=3). The head and neck patients all underwent laryngectomy and chemoradiotherapy and developed TEF secondarily.
line 150-151: these 2 patients didn't receive any reconstruction, for this reason should be removed from the study
=> Prior to submission, we discussed this topic intensively. We believe, that the option of "no treatment" should always be considered and is therefore an important part in any reconstructive algorithm. Therefore, we included this option in our treatment algorithm and would prefer to present these 2 patients in the manuscript. We highlighted this option in the discussion section to explain, why these two patients were included in the study .
"This algorithm also includes the option of “no-reconstruction”, whenever the physical status of the patient or the underlying disease do not allow extended surgical treatment. In these patients, the focus should be to do no harm and the relatively high perioperative risk has to be weighed up against the potential benefit."
line 155: about sternocleoidomastoidues flap, in which case has been used this flap? Is patient who received neck dissection? During neck dissection all pedicles of the muscle are resected and the it could result not vascularized, in these cases the procedure could become extremely hazardous (please explain this risk).
- The sternocleidomastoideus flap was applied in only one patient primarily, a second patient underwent a sternocleidomastoideus flap after primary reconstruction using an ICM-flap. Both patients suffered from esophageal cancers at the cervico-pharyngeal junction and therefore did not require neck dissection. We modified the text to the „treatment algorithm derived from the experience“ and added a sentence, that sternocleidomastoideus flaps may ba hazardous after neck dissections.
- “Patients suffering from extrathoracic TEF were treated using pedicled flap reconstructions as a first line treatment. Here, larger fistulae were reconstructed using a pectoralis major flap, whereas smaller lesions were planned for perforator flap reconstructions or a sternocleidomastoideus turnover, when its vascularity was not compromised by bilateral neck dissections.”
Line 159-161: “Another failure after pectoralis major reconstruction was finally salvaged using a combined ALT and anteromedial thigh (AMT) flap after a free ALT flap was lost due to persistent infection” this sentence is not clear and could be better built (did this patient receive 3 procedures?)
- We agree, that this sentence is hard to understand and may even be misleading. Therefore we modified the manuscript.
- „Here, one free jejunum flap was performed after insufficient reconstruction using a pectoralis major flap (PM). Another patient after PM reconstruction required a second reconstructive attempt using an ALT-Flap. Unfortunately this flap was also lost due to persistent infection, before TEF-closure was finally performed by a combined ALT and anteromedial thigh (AMT). One more case after discontinuity….
Punctuation in the legends is not correct, commas are used instead of semicolons and vice versa
- The manuscript was modified accordingly
Line 231: Authors state that first line in extrathoracic fistula their policy is to use pedicle flap, but all pedicle flap in cervical or cervical-thoracic fistula required second or third stage. How do the authors explain this fact, is radiotherapy much more impacting in head and neck area? under this light would be wiser to use free flap upfront in this area when possible (not in vessels depleted neck), add a sentence or in the algorithm please.
- This is a really important fact. We agree, that pedicled flaps showed higher complication rates in our series in head and neck patients. However, our overall experience in other head and neck cancer patients is much less disappointing. Therefore, we did not feel confident to change the algorithm due to the small number of patients. The literature comparing pedicled and free flaps (e.g. reference 13) likewise shows no clear trend in favor or against pedicled flaps. Since pedicled flaps require less ressources, we decided to include pedicled flaps as a first line treatment in the algorithm.
- Since we believe that this rationale should be clear for the readers we added a corresponding sentence to the discussion.
- „… whereas an incidence of 9% was described after fasciocutaneous free flaps [13]. In our series, we included two PMMC-flaps that both failed and required free flaps for reconstruction. However, we still included pedicled PMMC-flaps in the treatment algorithm as a first line option since we have good experiences using this flap in other cancer cases, the number of patients is low and the literature does not show a clear trend towards or against pedicled versus free flaps. Perez-Smith and colleagues describe leakage rates of 17% ……
Line 253, 255 brackets not in the right place
- corrected
Line 262: Author included, even in the algorithm, a vascularized pericardial flap, but they didn’t use in this series. Please remove at list from the algorithm
- the pericarial patch is removed from the algorithm
Reviewer 3 Report
The manuscript is interesting and well written. Although it includes a small sample of enrolled patients, it correctly analyzes the variables that affect the closure of the tracheoesophageal fistula. Moreover different procedures were described. To improve the overall quality of the manuscript, minor corrections are needed:
Introduction
- line 69, radiotherapy used in cancer patients also exposes the patient to failure of the subsequent vocal rehabilitation treatment, especially in the case of the use of tracheoesophageal voice prosthesis. please cite Serra A, Spinato G, Spinato R, et al. Multicenter prospective crossover study on new prosthetic opportunities in post-laryngectomy voice rehabilitation. J Biol Regul Homeost Agents. 2017;31(3):803-809.
- line 85, a prophylactic therapeutic option in case of advanced carcinomas where large resections are necessary or in irradiated tissues is the use of salivary Montgomery prostheses in order to prevent the formation of salivary fistulas and please cite Sevilla García MA, Suárez Fente V, Rodrigo Tapia JP, Llorente Pendás JL. El tubo de derivación salival de montgomery: una solución sencilla para las fístulas faringocutáneas [Montgomery salivary bypass tube: a simple solution for pharyngocutaneous fistulas]. Acta Otorrinolaringol Esp. 2006;57(10):467-470. doi:10.1016/s0001-6519(06)78750-7
Methods
- line 128, specify why PET was performed
- line 131, how do you choose between the procedures listed?
- line 146, specify the cases
- line 158, list the average number of procedure performed
- line 177, clasify why ''if possible''
Author Response
Introduction
line 69, radiotherapy used in cancer patients also exposes the patient to failure of the subsequent vocal rehabilitation treatment, especially in the case of the use of tracheoesophageal voice prosthesis. please cite Serra A, Spinato G, Spinato R, et al. Multicenter prospective crossover study on new prosthetic opportunities in post-laryngectomy voice rehabilitation. J Biol Regul Homeost Agents. 2017;31(3):803-809.
- Reference included in the manuscript and reference list.
line 85, a prophylactic therapeutic option in case of advanced carcinomas where large resections are necessary or in irradiated tissues is the use of salivary Montgomery prostheses in order to prevent the formation of salivary fistulas and please cite Sevilla García MA, Suárez Fente V, Rodrigo Tapia JP, Llorente Pendás JL. El tubo de derivación salival de montgomery: una solución sencilla para las fístulas faringocutáneas [Montgomery salivary bypass tube: a simple solution for pharyngocutaneous fistulas]. Acta Otorrinolaringol Esp. 2006;57(10):467-470. doi:10.1016/s0001-6519(06)78750-7
- Reference included in the manuscript and reference list.
Methods
line 128, specify why PET was performed
- A PET was performed when recommended by the multidisciplinary tumor board.
- This information was added to the manuscript.
- „Positron emission tomography (PET) was applied when recommended by the multidisciplinary tumor board.” .
line 131, how do you choose between the procedures listed?
- These diagnostic tools were applied according to the fistula localization.
- This information was added to the text
- “The digestive tract was evaluated using pharyngoscopy, laryngoscopy or esophago-gastro-duodenoscopy according to the localization of the fistula that was observed radiologically.
line 146, specify the cases
- An explanation was added to the manuscript
- „Prolonged side selective ventilation was performed in patients with bronchial fistulation to facilitate flap healing.“
line 158, list the average number of procedure performed
- this information was added
- Five patients required more than one procedure for reconstruction (average number of procedure: 2.2).
line 177, clasify why ''if possible''
- We agree, that this sentence is not precise. ICG-angiography was applied in all patients.
- Indocyanine green angiography was used to ensure and control vascular perfusion of free and pedicled flaps as well as gastrointensinal anastomoses [28-30].